# DUAL-MODE ASR: UNIFY AND IMPROVE STREAMING ASR WITH FULL-CONTEXT MODELING

**Jiahui Yu**[1]    **Wei Han**[1†]    **Anmol Gulati**[1†]    **Chung-Cheng Chiu**[1]    **Bo Li**[2]

**Tara N. Sainath**[2]    **Yonghui Wu**[1]    **Ruoming Pang**[1]

[1]Google Brain    [2]Google LLC

{jiahuiyu, rpang}@google.com

## ABSTRACT

*Streaming* automatic speech recognition (ASR) aims to emit each hypothesized word as quickly and accurately as possible, while *full-context* ASR waits for the completion of a full speech utterance before emitting completed hypotheses. In this work, we propose a unified framework, *Dual-mode ASR*, to train a single end-to-end ASR model with shared weights for both streaming and full-context speech recognition. We show that the latency and accuracy of streaming ASR significantly benefit from *weight sharing* and *joint training* of full-context ASR, especially with *inplace knowledge distillation* during the training. The Dual-mode ASR framework can be applied to recent state-of-the-art convolution-based and transformer-based ASR networks. We present extensive experiments with two state-of-the-art ASR networks, ContextNet and Conformer, on two datasets, a widely used public dataset LibriSpeech and a large-scale dataset MultiDomain. Experiments and ablation studies demonstrate that Dual-mode ASR not only simplifies the workflow of training and deploying streaming and full-context ASR models, but also significantly improves both emission latency and recognition accuracy of streaming ASR. With Dual-mode ASR, we achieve new state-of-the-art streaming ASR results on both LibriSpeech and MultiDomain in terms of accuracy and latency.

## 1 INTRODUCTION

"Ok Google. Hey Siri. Hi Alexa." have featured a massive boom of smart speakers in recent years, unveiling a trend towards ubiquitous and ambient Artificial Intelligence (AI) for better daily lives. As the communication bridge between human and machine, low-latency streaming ASR (*a.k.a.*, online ASR) is of central importance, whose goal is to emit each hypothesized word as quickly and accurately as possible on the fly as they are spoken. On the other hand, there are some scenarios where full-context ASR (*a.k.a.*, offline ASR) is sufficient, for example, offline video captioning on video-sharing platforms. While low-latency streaming ASR is generally preferred in most of the speech recognition scenarios, it often has worse prediction accuracy as measured in Word Error Rate (WER), due to the lack of future context compared with full-context ASR. Improving both WER and emission latency has been shown to be highly challenging (He et al., 2019; Li et al., 2020a; Sainath et al., 2020) in streaming ASR systems.

Since the acoustic, pronunciation, and language model (AM, PM, and LM) of a conventional ASR system have been evolved into a single end-to-end (E2E) all-neural network, modern streaming and full-context ASR models share most of the neural architectures and training recipes in common, such as, Mel-spectrogram inputs, data augmentations, neural network meta-architectures, training objectives, model regularization techniques and decoding methods. The most significant difference is that *streaming ASR encoders are auto-regressive models*, with the prediction of the current timestep conditioned on previous ones (no future context is permitted). Specifically, let $x$ and $y$ be the input and output sequence, $t$ as frame index, $T$ as total length of frames. Streaming ASR encoders model the output $y_t$ as a function of input $x_{1:t}$ while full-context ASR encoders model the output $y_t$ as a function of input $x_{1:T}$. Streaming ASR encoders can be built with uni-directional LSTMs, causal convolution and left-context attention layers in streaming ASR encoders (Chiu & Raffel, 2018; Fan et al., 2018; Han et al., 2020; Gulati et al., 2020; Huang et al., 2020; Moritz et al., 2020; Miao

---
†equal contribution

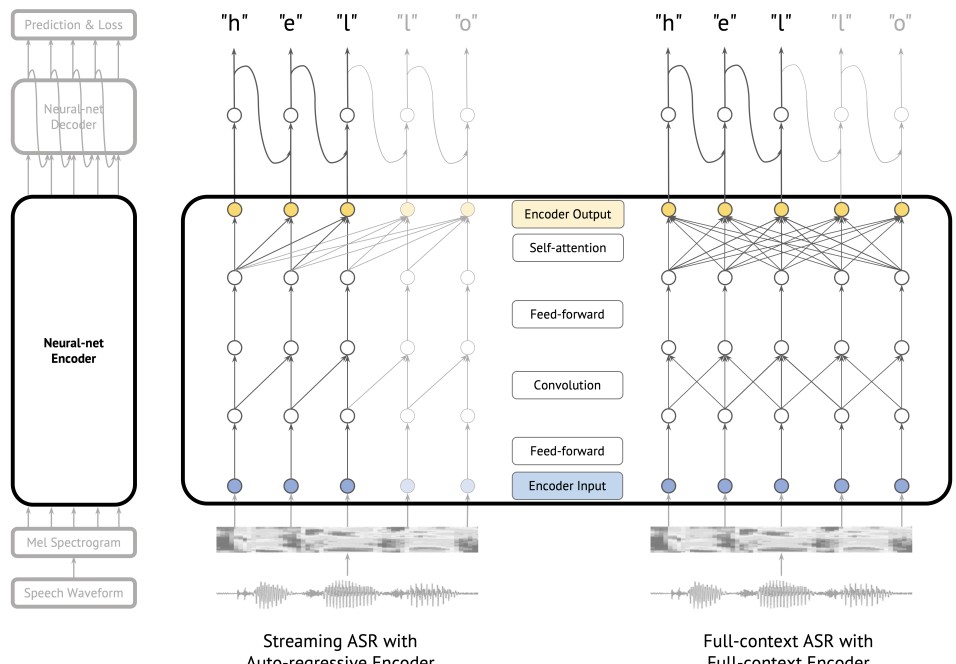

Figure 1: A simplified illustration of the similarity and difference between Streaming ASR and Full-context ASR networks. Modern end-to-end streaming and full-context ASR models share most of the neural architectures and training recipes in common, with the most significant difference in the **ASR encoder (highlighted)**. Streaming ASR encoders are auto-regressive models, with each prediction of the current timestep conditioned on previous ones (no future context). We show examples of feed-forward layer, convolution layer and self-attention layer in the encoder of streaming and full-context ASR respectively. With Dual-mode ASR, we unify them without parameters overhead.

et al., 2020; Tsunoo et al., 2020; Zhang et al., 2020; Yeh et al., 2019). Recurrent Neural Network Transducers (RNN-T) (Graves, 2012) are commonly used as the decoder in both streaming and full-context models, which predicts the token of the current input frame based on all previous tokens using uni-directional recurrent layers. Figure 1 illustrates a simplified example of the similarity and difference between streaming and full-context ASR models with E2E neural networks.

Albeit the similarities, streaming and full-context ASR models are usually developed, trained, and deployed separately. In this work, we propose *Dual-mode ASR*, a framework to unify streaming and full-context speech recognition networks with shared weights. Dual-mode ASR comes with many immediate benefits, including reduced model download and storage on devices and simplified development and deployment workflows. To accomplish this goal, we first introduce *Dual-mode Encoders*, which can run in both streaming mode and full-context mode. Dual-mode encoders are designed to reuse the same set of model weights for both modes with zero or near-zero parameters overhead. We propose the design principles of a dual-mode encoder and show examples on how to design dual-mode convolution, dual-mode pooling, and dual-mode attention layers. We also investigate into different training algorithms for Dual-mode ASR, specifically, randomly sampled training and joint training. We show that joint training significantly outperforms randomly sampled training in terms of model quality and training stability. Moreover, motivated by *Inplace Knowledge Distillation* (Yu & Huang, 2019b) in which a large model is used to supervise a small model, we propose to *distill knowledge from the full-context mode (teacher) into the streaming mode (student) on the fly* during the training within the same Dual-mode ASR model, by encouraging consistency of the predicted token probabilities.

We demonstrate that the emission latency and prediction accuracy of streaming ASR significantly benefit from *weight sharing* and *joint training* of its full-context mode, especially with *inplace knowledge distillation* during the training. We present extensive experiments with two state-of-the-art ASR networks, convolution-based ContextNet (Han et al., 2020) and conv-transformer hybrid Conformer (Gulati et al., 2020), on two datasets, a widely used public dataset LibriSpeech (Panay-

otov et al., 2015) (970 hours of English reading speech) and a large-scale dataset MultiDomain (Narayanan et al., 2018) (413,000 hours speech of a mixture across multiple domains including Voice Search, Farfield Speech, YouTube and Meetings). For each proposed technique, we also present ablation study and analysis to demonstrate and understand the effectiveness. With Dual-mode ASR, we achieve new state-of-the-art streaming ASR results on both LibriSpeech and MultiDomain in terms of accuracy and latency.

## 2 RELATED WORK

**Streaming ASR Networks.** There has been a growing interest in building streaming ASR systems based on E2E Recurrent Neural Network Transducers (RNN-T) (Graves, 2012). Compared with sequence-to-sequence models (Chorowski et al., 2014; 2015; Chorowski & Jaitly, 2016; Bahdanau et al., 2016; Chan et al., 2016), RNN-T models are naturally *streamable* and have shown great potentials for low-latency streaming ASR (Chang et al., 2019; He et al., 2019; Tsunoo et al., 2019; Sainath et al., 2019; Shen et al., 2019; Li et al., 2020a;b; Sainath et al., 2020; Huang et al., 2020; Moritz et al., 2020; Narayanan et al., 2020). In this work, we mainly focus on RNN-T based models. He et al. specifically studied how to optimize the RNN-T streaming ASR model for mobile devices, and proposed a bag of techniques including using layer normalization and large batch size to stabilize training; using word-piece targets (Wu et al., 2016); using a time-reduction layer to speed up training and inference; quantizing network parameters to reduce memory footprint and speed up computation; applying shallow-fusion to bias towards user-specific context. To support streaming modeling in E2E ASR models, various efforts have also been made by modifying attention-based models such as monotonic attention (Raffel et al., 2017; Chiu & Raffel, 2017; Fan et al., 2018; Arivazhagan et al., 2019), GMM attention (Graves, 2013; Chiu et al., 2019), triggered attention (TA) (Moritz et al., 2019), Scout Network (Wang et al., 2020), and approaches that segment encoder output into non-overlapping chunks (Jaitly et al., 2016; Tsunoo et al., 2020). Tsunoo et al. also applied knowledge distillation from the non-streaming model to the streaming model, but their streaming and non-streaming models do not share weights and are trained separately.

To improve the latency of RNN-T streaming models, Li et al. investigated additional early and late penalties on Endpointer prediction (Chang et al., 2019) to reduce the emission latency, and employed the minimum word error rate (MWER) training (Prabhavalkar et al., 2018) to remedy accuracy degradation. Sainath et al. further proposed to improve quality by using two-pass models (Sainath et al., 2019), *i.e.*, a second-pass LAS-based rescore model on top of the hypotheses from first-pass RNN-T streaming output. More recently, Li et al. proposed parallel rescoring by replacing LSTMs with Transformers (Vaswani et al., 2017) in rescoring models. Chang et al. further proposed Prefetching to reduce system latency by submitting partial recognition results for subsequent processing such as obtaining assistant server responses or second-pass rescoring before the recognition result is finalized. Unlike these approaches, our work explores the unification of streaming and full-context ASR networks, thus can be generally applied as an add-on technique without requiring extra runtime support during inference.

**Weight Sharing for Multi-tasking.** Sharing model weights of a deep neural network for multiple tasks has been widely explored in the literature to reduce overall model sizes. In the broadest sense, tasks can refer to different objectives or same objective but different settings, ranging from natural language processing and speech recognition to computer vision and reinforcement learning. In speech recognition, Kannan et al. employed a single ASR network for multilingual ASR, and showed accuracy improvements over monolingual ASR systems. Wu et al. proposed dynamic sparsity neural networks (DSNN) for speech recognition on mobile devices with resource constraints. A single trained DSNN (Wu et al., 2020) can transform into multiple networks of different sparsities for adaptive inference in real-time. Chang et al. trained a single RNN-T model with LSTMs (Hochreiter & Schmidhuber, 1997) for Joint Endpointing (*i.e.*, predicting both recognition tokens and the end of an utterance transcription) in streaming ASR systems. Moreover, Watanabe et al. proposed a hybrid CTC and attention architecture for ASR based on multi-objective learning to eliminate the use of linguistic resources.

Another related research work in Computer Vision is Slimmable Neural Networks (Yu et al., 2018; Yu & Huang, 2019a;b; Yu et al., 2020). Yu et al. proposed an approach to train a single neural network running at different widths, permitting instant and adaptive accuracy efficiency trade-offs at

runtime. We also adapt the training rules introduced in slimmable networks, that is, using independent normalization layers for different sub-networks (tasks) as conditional parameters and using the prediction of teacher network to supervise student network as inplace distillation during the training. Unlike slimmable networks in which a large model is used to supervise a small model, we propose to distill the knowledge from full-context mode (teacher) into streaming mode (student) on the fly within the same Dual-mode ASR model.

**Knowledge Distillation.** Hinton et al. explored a simple method to "transfer" knowledge from a teacher neural network to a student neural network by enforcing their predictions to be close measured by KL-divergence, $\ell_1$ or $\ell_2$ distance. It is shown that such distillation method is effective to compress neural networks (Yu & Huang, 2019b), accelerate training (Chen et al., 2015), improve robustness (Carlini & Wagner, 2017; Papernot et al., 2016), estimate model uncertainty (Blundell et al., 2015) and transfer learned domain to other domains (Tzeng et al., 2015).

## 3 DUAL-MODE ASR

Most neural sequence transduction networks for ASR have an encoder-decoder structure (Graves, 2012; Sainath et al., 2020; He et al., 2019; Li et al., 2020a), as shown in Figure 1. Without loss of generality, here we discuss how to design Dual-mode ASR networks under the most commonly used RNN-T model (Graves, 2012). In RNN-T models, we first extract mel-spectrogram feature from input speech waveform. The Mel-spectrogram feature is then fed into a neural-net encoder, which usually consists of feed-forward layers, RNN/LSTM layers, convolution layers, attention layers, pooling (time-reduction) layers, and residual or dense connections. In neural-net encoders, streaming ASR model requires all components to be auto-regressive, whereas full-context ASR model has no such requirement. The ASR decoder then predicts the token of current frame based on the output from the encoder and previous predicted tokens (inference) or target tokens (training with teacher forcing (Williams & Zipser, 1989)). The decoder is commonly an auto-regressive model in both streaming and full-context ASR models, thus is *fully shared* in Dual-mode ASR. The prediction from decoder is finally used either in decoding algorithm during inference (*e.g.*, beam search) or learning algorithm during training (*e.g.*, RNN-T loss).

As discussed above and shown in Figure 1, it becomes clear that the major difference between streaming and full-context ASR models is in the *neural-net encoder*. In the following, we will first discuss the design principles of *dual-mode encoder* to support both streaming and full-context ASR. We provide examples including dual-mode convolution, dual-mode average pooling, and dual-mode attention layers, which are widely used in the state-of-the-art ASR networks ContextNet (Han et al., 2020) and Conformer (Gulati et al., 2020). We will then discuss the training algorithm of Dual-mode ASR networks including joint training and inplace knowledge distillation.

### 3.1 DUAL-MODE ENCODER

Unifying streaming and full-context ASR models requires two design principles of *Dual-mode Encoder*:

1. Each layer in a dual-mode encoder should be either dual-mode or streaming (*a.k.a.*, causal). Since streaming encoder has to be auto-regressive which prohibits any future context, any full-context (*a.k.a.*, non-causal) layer violates this constraint.

2. The design of a dual-mode layer should not introduce significant amount of additional parameters, compared with its streaming model. We aim at supporting full-context ASR on top of the streaming model with near-zero parameters overhead.

We show examples below by applying the above two design principles to ContextNet (Han et al., 2020) and Conformer (Gulati et al., 2020), in which the encoders are composed of pointwise operators (feed-forward net, residual connections, activation layers, striding, dropout, *etc*.), convolution, average pooling, self-attention and normalization layers.

**Pointwise operators are naturally dual-mode layers.** Neural network layers that connect input and output neurons within each timestep (no across-connections among different timesteps) are often referred as pointwise operators (Chollet, 2017), including feed-forward layers (*a.k.a.*, fully-connected layers or $1 \times 1$ convolution layers), activation layers (*e.g.*, ReLU, Swish (Ramachandran

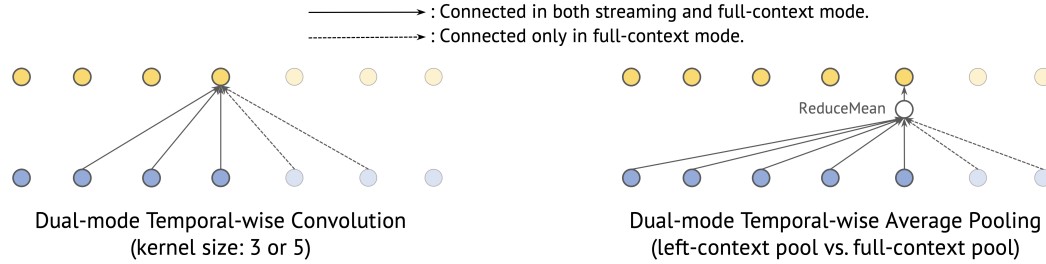

Figure 2: Dual-mode convolution and average pooling layer for Dual-mode ASR.

et al., 2017)), residual and dense connections (He et al., 2016; Huang et al., 2017), striding layers, dropout layers (Srivastava et al., 2014) and element-wise multiplications. As there is no information propagation through time, pointwise operators are naturally dual-mode layers and can be directly used in Dual-mode ASR encoders.

**Dual-mode Convolution.** Convolution layers, however, convolve feature across its neighbor timesteps within a fixed window (*e.g.*, kernel size is 3, 5, or larger), and has been widely used in sequence modeling (Gehring et al., 2017; Han et al., 2020; Gulati et al., 2020). In conv-based streaming ASR models, causal convolution layers (Oord et al., 2016) are used where the convolution window is biased to the left (self-included). As shown in Figure 2 on the left, to support both streaming and full-context modes with shared weights, we first construct a normal symmetric convolution of kernel size $k$ which will be applied in full-context mode. Then we mimic the causal convolution of kernel size $(k + 1)/2$ by constructing a Boolean mask and multiplying with the full-context convolution kernel before applying the actual convolution of streaming mode in Dual-mode ASR encoders.

The design of dual-mode convolution introduces $(k - 1)/2$ additional parameters to support full-context convolution ($k$) compared with streaming convolution ($(k+1)/2$). However, we note that in convolution-based models, these temporal-wise convolution layers only take a tiny amount of total model size and most of the weights are on $1 \times 1$ convolution layers which are fully shared pointwise operators. For example, in ContextNet (Han et al., 2020), temporal-wise convolution has less than 1% of total model size, thus parameters overhead is negligible.

**Dual-mode Average Pooling.** Squeeze-and-excitation (Hu et al., 2018) (SE) modules are used in ContextNet to enhance the global context encoding. Each SE module is a sequential stack of average pooling (through time) layer, feed-forward layer, activation layer, another feed-forward layer and elementwise multiplication. To support both modes, dual-mode average pooling layer is used as shown in Figure 2 on the right. Dual-mode average pooling layer is parameter-free thus does not introduce additional model parameters. It also trains in parallel in streaming mode, easily implemented with "cumsum" function in both TensorFlow and PyTorch.

**Dual-mode Self-attention.** Self-attention (*a.k.a.* intra-attention) is an attention mechanism weighting different positions of a single sequence in order to compute a representation of the same sequence. It is heavily used in Conformer (Gulati et al., 2020) ASR networks. The attention layer itself is parameter-free (projection layers before attention are fully shared), and is composed of matrix multiplication of the key and the query, followed by softmax over keys, before another matrix multiplication with the value. As shown in Figure 3, in dual-mode attention layer, the softmax is performed on the left context only in streaming mode (rectangle with solid line), compared with the full-context mode (rectangle with dash line). We find this simple form of dual-mode self-attention works well in practice.

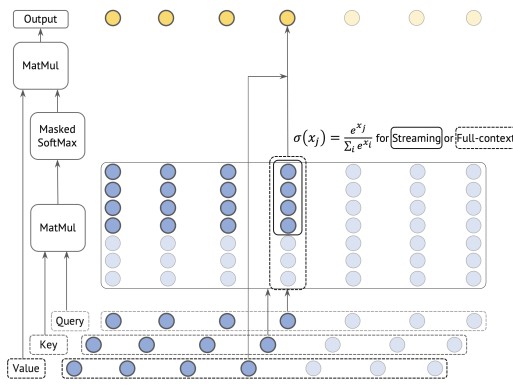

Figure 3: Dual-mode self-attention layer.

---

**Algorithm 1** Pseudocode of training Dual-mode ASR networks.

---

```
# Requires: data_loader; context manager with support of mode switching by network.mode();
    dual_mode_network with support of running both modes under context manager;

for x, y in data_loader: # Load a minibatch of speech input x and text label y.
    with dual_mode_network.mode('fullcontext'): # Switch context to 'fullcontext' mode.
        # Compute full-context prediction given speech input x and text label y.
        fullcontext_pred = dual_mode_network.forward_encoder_decoder(x, y)
        # Compute RNN-T loss of full-context mode.
        fullcontext_loss = rnnt_loss(fullcontext_pred, y)

    with dual_mode_network.mode('streaming'): # Switch context to 'streaming' mode.
        # Compute streaming prediction given speech input x and text label y.
        streaming_pred = dual_mode_network.forward_encoder_decoder(x, y)
        # Compute RNN-T loss of streaming mode.
        streaming_loss = rnnt_loss(streaming_pred, y)

    # Add inplace knowledge distillation loss (full-context prediction as teacher).
    distill_loss = inplace_distill_loss(streaming_pred, stop_gradient(fullcontext_pred))

    # Compute total loss as a sum of full-context, streaming and distillation losses.
    loss = fullcontext_loss + streaming_loss + distill_loss
    loss.backward() # Update weights.
```

---

**Dual-mode Normalization.** Moreover, following Yu et al. (2018), we also find the normalization statistics like means and variances are different in streaming and full-context modes. Thus, for normalization layers including BatchNorm (Ioffe & Szegedy, 2015) and LayerNorm (Ba et al., 2016) in Dual-mode ContextNet and Dual-mode Conformer, we instantiate two separate norm layers dedicated to streaming and full-context mode respectively.

## 3.2 TRAINING DUAL-MODE ASR NETWORKS

The training algorithm of Dual-mode ASR networks is outlined in Algorithm 1. In this section, we discuss two important training techniques: joint training and inplace knowledge distillation.

**Joint Training.** To train Dual-mode ASR networks, given a batch of data in each training iteration, we can either randomly sample one from two modes to train, or train both modes and aggregate their losses. In the former approach, referred as *randomly sampled training*, we can control the importance of streaming and full-context modes by setting different sampling probabilities during training. In the latter approach, referred as *joint training*, importance can also be controlled by assigning different loss weights to balance streaming and full-context modes. Empirically we find joint training leads to better model qualities overall thus is adopted in all of our experiments. We will show an ablation study comparing randomly sampled training and joint training. In all of our experiments, we treat streaming and full-context mode to be equally important by assigning equal importance during training.

**Inplace Knowledge Distillation.** Additionally we propose to distill knowledge from the full-context mode (teacher) into the streaming mode (student) on the fly within the same Dual-mode ASR model, by encouraging consistency of the predicted token probabilities. Since in each iteration we always compute predictions of both modes, the teacher prediction comes for free (no additional computation or memory cost), as shown in Algorithm 1. We use the efficient knowledge distillation introduced by Panchapagesan et al., which is based on the KL-divergence between full-context and streaming over the probability of three parts: $P_{label}$, $P_{blank}$ and $1 - P_{label} - P_{blank}$. We note that the prediction of full-context mode (teacher) usually has lower latency (since it has no incentive to delay its output), thus we can control the target emission latency of streaming mode (student) by shifting the prediction of full-context mode, before applying distillation loss. We do a small-scale hyper-parameter sweep from -2 to 2 frames to shift for ContextNet and Conformer in our experiments.

## 4 EXPERIMENTS

## 4.1 MAIN RESULT

**Measuring Latency.** Latency measurement is itself challenging for streaming ASR systems. Motivated by Prefetching (Chang et al., 2020) technique, we measure latency as the difference of two

timestamps: 1) when the last token is emitted in the finalized recognition result; 2) the end of the speech when a user finishes speaking. We find this is especially descriptive of user experience in real-world ASR applications like Voice Search. ASR models that capture stronger contexts can emit the full hypothesis even before they are spoken, leading to a **negative latency**. Moreover, instead of naively averaging latency over all utterances, we report both median and 90th percentile of all utterances in test set, denoted as **Latency@50** and **Latency@90**, to better characterize latency by excluding outlier utterances. To evaluate the model quality, we report WER only for full-context models and both WER and latency for streaming models (full-context latency is meaningless).

**Datasets.** We conduct our experiments on two datasets: a public widely used dataset LibriSpeech (Panayotov et al., 2015) (1,000 hours of English reading speech) and a large-scale dataset MultiDomain (413,000 hours speech, 287 million utterances of a mixture across multiple domains including Voice Search, YouTube, and Meetings). Table 1 summarizes the information and statistics of two datasets. For LibriSpeech, we report our evaluation results on TestClean and TestOther (noisy) sets and compare with other published baselines. For MultiDomain, we report our evaluation results on Voice Search test set and compare with our reproduced baselines. For fair comparisons, on each dataset we train and report our models and baselines with the same settings (number of training iterations, hyper-parameters, optimizer, regularization, *etc*.). We note that these hyper-parameters are inherited from previous work Han et al. (2020); Gulati et al. (2020) and not specifically tuned for our dual-mode models.

Table 1: Summary of datasets we used in our experiments.

| Dataset Name | # Hours | # Utterances | Speech Domain |
|---|---|---|---|
| LibriSpeech (Panayotov et al., 2015) | $\sim 970$ | $\sim 281,000$ | Single domain of English reading speech. |
| MultiDomain (Narayanan et al., 2018) | $\sim 413,000$ | $\sim 287,000,000$ | Multiple domains including: Voice Search, Farfield Speech, YouTube and Meetings. |

**ASR Networks.** We use two recent state-of-the-art ASR networks to demonstrate the effectiveness of our proposed methods, ContextNet (Han et al., 2020) and Conformer (Gulati et al., 2020). The encoder of ContextNet is based on depthwise-separable convolution (Chollet, 2017) and squeeze-and-excitation modules (Hu et al., 2018). In depthwise-separable convolution of Dual-mode ContextNet, the weights of $1 \times 1$ convolutions are fully shared between streaming and full-context mode, whereas for temporal-wise convolution we follow the design of Dual-mode Convolution proposed in Section 3.1. Note that in ContextNet, temporal-wise convolutions only take less than 1% of the model size thus the parameters overhead of full-context mode is negligible compared with steaming mode. In squeeze-and-excitation modules, we use dual-mode average pooling layers (Section 3.1) to support both streaming and full-context mode without additional parameters.

Conformer (Gulati et al., 2020) combines convolution and transformer to model both local and global dependencies of speech sequences in a parameter-efficient way. In Dual-mode Conformer, we replace all convolution and transformer layers with their dual-mode correspondents (Section 3.1). Moreover, for normalization layers including BatchNorm (Ioffe & Szegedy, 2015) and LayerNorm (Ba et al., 2016) in Dual-mode ContextNet and Dual-mode Conformer, we instantiate two separate norm layers for streaming and full-context mode respectively.

**Training Details and Results.** We train our models exactly following our baselines ContextNet (Han et al., 2020) and Conformer (Gulati et al., 2020), using Adam optimizer (Kingma & Ba, 2014), SpecAugment (Park et al., 2019) and a transformer learning rate schedule (Vaswani et al., 2017) with warm-up (Goyal et al., 2017). Our main results are summarized in Table 2 and Table 3. We also add a streaming ContextNet Look-ahead baseline (6 frames, 10ms per frame, totally 60ms look-ahead latency) in Table 3 by padding additional frames at the end of the input utterances. As shown in the tables, the streaming mode in Dual-mode ASR models has significantly better latency and similar or higher WER results, surpassing other baselines including conventional models, LSTM-based transducers (Sainath et al., 2020), transformer-transducers (Zhang et al., 2020) and some others.

Table 2: Summary of our results on MultiDomain dataset (Narayanan et al., 2018). We report WER on Voice Search test set. Compared with standalone ContextNet and Conformer models, Dual-mode ASR models have slightly higher accuracy and much better streaming latency. ASR models that capture stronger contexts can emit the full hypothesis even slightly before they are spoken, leading to a *negative latency*.

| Method | Mode | # Params (M) | VS Test WER(%) | Latency@50 (ms) | Latency@90 (ms) |
|---|---|---|---|---|---|
| ContextNet | Full-context | 133 | 5.1 | —— | —— |
| Conformer | Full-context | 142 | 5.2 | —— | —— |
| LSTM (Sainath et al., 2020) | Streaming | 179 | 6.4 | 190 | 350 |
| ContextNet (Han et al., 2020) | Streaming | 133 | 6.1 | 160 | 310 |
| Conformer (Gulati et al., 2020) | Streaming | 142 | 6.1 | 160 | 300 |
| Dual-mode ContextNet | Full-context Streaming | 133 | 4.9 6.0 (-0.1) | —— 10 (-150) | —— 220 (-90) |
| Dual-mode Conformer | Full-context Streaming | 142 | 5.0 6.0 (-0.1) | —— -50 (-210) | —— 130 (-170) |

Table 3: Summary of our results on Librispeech dataset (Panayotov et al., 2015). We report WER on TestClean and TestOther (noisy) set. Compared with standalone ContextNet and Conformer models, Dual-mode ASR models have both higher accuracy in average and better streaming latency.

| Method | Mode | # Params (M) | Test Clean/Other WER(%) | Latency@50 (ms) | Latency@90 (ms) |
|---|---|---|---|---|---|
| LSTM-LAS | Full-context | 360 | 2.6 / 6.0 | —— | —— |
| QuartzNet-CTC | Full-context | 19 | 3.9 / 11.3 | —— | —— |
| Transformer | Full-context | 29 | 3.1 / 7.3 | —— | —— |
| Transformer | Full-context | 139 | 2.4 / 5.6 | —— | —— |
| ContextNet | Full-context | 31.4 | 2.4 / 5.4 | —— | —— |
| Conformer | Full-context | 30.7 | 2.3 / 5.0 | —— | —— |
| Transformer | Streaming | 18.9 | 5.0 / 11.6 | 80 | 190 |
| ContextNet | Streaming | 31.4 | 4.5 / 10.0 | 70 | 270 |
| Conformer | Streaming | 30.7 | 4.6 / 9.9 | 140 | 280 |
| ContextNet Look-ahead | Streaming | 31.4 | 4.1 / 9.0 | 150 | 420 |
| Dual-mode Transformer | Full-context Streaming | 29 | 3.1 / 7.9 4.4 (-0.6) / 11.5 (-0.1) | —— -50 (-130) | —— 30 (-160) |
| Dual-mode ContextNet | Full-context Streaming | 31.8 | 2.3 / 5.3 3.9 (-0.6) / 8.5 (-1.5) | —— 40 (-30) | —— 160 (-110) |
| Dual-mode Conformer | Full-context Streaming | 30.7 | 2.5 / 5.9 3.7 (-0.9) / 9.2 (-0.7) | —— 10 (-130) | —— 90 (-190) |

## 4.2 ABLATION STUDY

In this section, we perform various ablation studies to support and understand the effectiveness of each technique in Dual-mode ASR. We train Dual-mode ContextNet on LibriSpeech training set with exactly same settings and report WER, Latency@50 and Latency@90 on TestOther set of streaming mode. We specifically study three techniques and their combinations including *weight sharing*, *joint training* and *inplace knowledge distillation* during the training.

During the training we distill knowledge from full-context mode (teacher) into streaming mode (student) on the fly within the same dual-mode model. Inplace distillation during the training comes for free as shown in training Algorithm 1. But what if we simply share weights and jointly train them without distillation? As shown in the second row of Table 4, the model without inplace distillation during the training has worse results compared to the baseline.

Given a batch of data for each training iteration, we train both modes and aggregate their losses. We also show results of *randomly sampled training* in the third row of Table 4, which leads to even worse performance. Note that with randomly sampled training, we cannot apply inplace distillation

easily either because in each training iteration there is only one prediction from either streaming mode or full-context mode.

Weight sharing reduces the model size which is one of the major motivation of Dual-mode ASR. However, what if we simply train two individual models and use knowledge distillation with full-context model as the teacher? As shown in the last row of Table 4, the results are better than other ablation but still worse than the Dual-mode ASR baseline. It might indicate that weight sharing itself encourages learning better deep representation for streaming ASR. Weight sharing has been shown empirically to improve Multilingual ASR (Kannan et al., 2019), Model Pruning (Wu et al., 2020), Endpointing (Hochreiter & Schmidhuber, 1997) and some Computer Vision problems (Yu et al., 2018) and this intriguing property need to be studied in more details as a future work.

Table 4: Ablation studies of weight sharing, joint training and inplace distillation. We report WER on TestOther (noisy) set (Panayotov et al., 2015) using ContextNet with same training settings.

| Weight Sharing | Joint Training | Inplace Distillation | TestOther WER(%) | Latency@50 (ms) | Latency@90 (ms) |
|---|---|---|---|---|---|
| ✔ | ✔ | ✔ | 8.5 | 40 | 160 |
| ✔ | ✔ | ✗ | 10.2 (+1.7) | 120 (+80) | 310 (+150) |
| ✔ | ✗ | ✗ | 10.6 (+2.1) | 90 (+50) | 290 (+130) |
| ✗ | ✔ | ✔ | 9.9 (+1.4) | 50 (+10) | 210 (+50) |

Further, we visualize the emission lattices of dual-mode ASR models trained with and without inplace knowledge distillation. We randomly sampled two audio sequences on LibriSpeech TestOther set and plotted their emission lattices of streaming mode in Figure 4. X-axis represents the speech input frames while Y-axis represents the text output labels (tokens). Figure 4 shows that with knowledge distillation from full-context mode in Dual-mode ASR, streaming mode emits faster and has much less latency, which is very critical for product datasets like MultiDomain presented in our work.

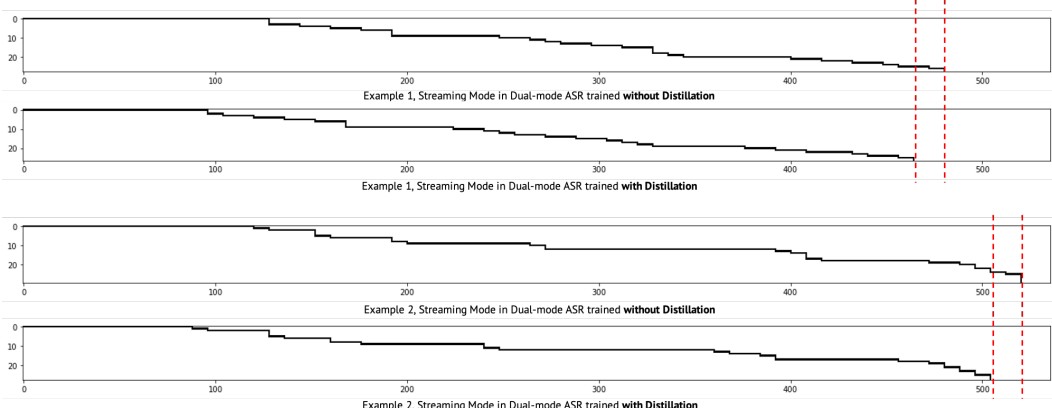

Figure 4: Two speech-text pair comparison of Dual-model ASR models trained with and without inplace distillation by visualization of their streaming emission lattices. X-axis represents the speech input frames while Y-axis represents the text output labels (tokens). Inplace distillation significantly reduces emission latency of streaming mode in Dual-mode ASR models which is critical in real-world applications.

## 5 CONCLUSION

In this work, we have proposed a unified framework, *Dual-mode ASR*, to unify and improve streaming ASR by joint full-context modeling. We hope our exploration will inspire streaming models in other fields such as simultaneous machine translation and video processing.

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
