# OpenReview forum: "Dual-mode ASR: Unify and Improve Streaming ASR with Full-context Modeling"
_ICLR.cc/2021/Conference — ICLR 2021 Poster_

### Official Review · AnonReviewer3 · 2020-10-25
**Review by AnonReviewer3**

**Rating:** 6
**Confidence:** 5

**Review:**

This submission proposes a framework for training online and offline ASR models. Experimental results suggest that at least on Librispeech this approach provides tangible benefits for online ASR models.

Quality: The quality of this submission suffers from (a) mostly verbal presentation (e.g. Figure 1 is a very inefficient way to show that prediction at time t is either dependent on past only input or all of the input) and (b) limited benefits observed on the challenging MultiDomain data set.  Regarding (a), you work would have been significantly stronger if you would have provided more technical descriptions of changes that you are proposing to ensure that the elements you are using are online "friendly". This might have helped you to link what you are proposing to other work done in the past and further strengthen your submission. Regarding (b), it seems that MultiDomain data set is very challenging or not enough tuning was performed to illustrate the benefit of your approach.

Clarity: The clarity of this submission suffers from a mostly verbal presentation of very technical operations.

Originality: This submission offers limited originality.

Significance: Given experimental results, the significance of this particular submission is minor.

Pros: This submission presents what I believe are generally useful techniques but the presentation is verbal rather than mathematical which makes establishing connections significantly harder than it needs to be. At least on one of the data sets the results appear to be good.

Cons: Technical elements are described in a very verbal fashion which may lead to misinterpretation. The results on more challenging MultiDomain data set do not make a convincing case that the proposed solution or its tuning is more generally useful.

Post author response stage: Given the response from the authors and the input from other reviewers I increased the score from 4 to 6.

---

> ### Author Response · Authors · 2020-11-24
> **Authors' Reply to AnonReviewer3**
>
> Thanks for your time reviewing and providing detailed suggestions for our manuscript! We have updated the manuscript according to your suggestions and addressed all concerns below.
>
> 1. Suffer from verbal presentation of very technical operations and inefficient Figure 1.
>
> Thanks for your comments. We have updated our manuscript to include more mathematical and formal presentation of the difference between streaming and non-streaming ASR models in Introduction.
>
> 2. Limited improvement on challenging MultiDomain dataset. Minor significance of results.
>
> First, we want to emphasize that our improvement to streaming emission latency is indeed huge on both LibriSpeech and MultiDomain. In addition, the LibriSpeech WER has been improved by more than 20% relative improvements, which is not “minor” in the literature. While the improvement of WER on MultiDomain appears to be limited, the reduction of latency is more critical for the real-world streaming ASR applications. Also, any improvement of WER should be considered as additional benefits as Dual-mode ASR now supports both streaming and non-streaming ASR within a single weight-shared model, which is itself useful---as confirmed by other reviewers.
>
> 3. Technique is generally useful.
>
> Thanks for your acknowledgement on the usefulness of the proposed Dual-mode ASR!

---

### Official Review · AnonReviewer4 · 2020-10-28
**A study that could benefit the ASR community a lot**

**Rating:** 7
**Confidence:** 4

**Review:**

This paper proposes an unified framework for both streaming and non-streaming ASR and the knowledge transfer between them. The results show that both latency and performance are improved. The benefit of training full-context and streaming together are two folds: 1) Current full-context and streaming ASR are trained separately. Since usually the performance of streaming ASR is inferior to the full-context version, the unified training scheme could enforce the model to fit both tasks well, thus could serve as some kind of regularization. 2) The weight sharing proposed in this paper could make it more efficient for deploying both streaming and non-streaming ASR at the same time.

Cons:
1. Though conformer and context achieved SOTA performance as full-context models, they are pretty new and not widely acknowledged and studied compared to Transformer. Adding results of Transformer could better support the claims and make the contribution in this paper more accessible to the community.
2. The source of improvement on latency is not well explained.
3. “It might indicate that weight sharing itself encourages learning better deep representation for streaming ASR.”, this claim should be further validated.

Question and comments:
1. Adding some related work on knowledge distillation could make it a more complete story.
2. Seems the statistics of MultiDomain dataset are not consistent in the paper (413,000 hours vs 163,000 hours)
3. "For fair comparisons, on each dataset we train and report our models and baselines with the same settings (number of training iterations, hyper-parameters, optimizer, regularization, etc.)." Could it be possible different models perform the best with different hyperparameter settings?
4. It could be interesting if the visualization of knowledge transferred from full-context model to streaming model is investigated.

---

> ### Author Response · Authors · 2020-11-24
> **Authors' Reply to AnonReviewer4**
>
> Thanks for your time reviewing and providing detailed suggestions for our manuscript! We have updated the manuscript according to your suggestions and addressed all concerns below.
>
> ## Part 1: General Questions
>
> Q: Transformer could be added.
>
> A: First, Conformer is a variant of Transformer (with additional convolution layers) and has performed better than pure transformers in our extensive studies. The experiments of Dual-mode Conformer is a demonstration of the effectiveness of dual-mode self-attention layers. Nevertheless, we have included our reproduced pure Transformer models and Dual-mode Transformer ASR model on the LibriSpeech dataset in Table 3. The results and conclusions are consistent with our Conformer models.
>
> Q: Explain the source of latency improvement.
>
> A: As shown in Ablation Study Table 4, as non-streaming ASR models do not have latency, we found joint training streaming ASR and non-streaming ASR together with weight sharing and distillation enforces streaming ASR models to intuitively “mimic” non-streaming models thus emit faster with lower emission latency. We have also added a visual comparison of Dual-mode ASR models with and without knowledge distillation to better show visually how the latency can be improved.
>
> Q: Validate the effectiveness of weight sharing.
>
> A: Weight sharing, as discussed in Related Work, has been shown empirically to improve Multilingual ASR, Model Pruning, Endpointing and some Computer Vision problems. We agree that further study on this intriguing property of weight sharing is necessary and we have updated the manuscript to study further in more details as future work.
>
>
> ## Part 2: Detailed Questions / Comments / Suggestions
>
> 1. Related work on knowledge distillation.
>
> Thanks for your suggestion! We have updated the manuscript and added the discussion in Related Work.
>
> 2. MultiDomain dataset are not consistent.
>
> Thanks for the catch! The new MultiDomain has been expanded to 413,000 hours on which we conducted our experiments. We have updated the manuscript to be consistent.
>
> 3. Could it be possible different models perform the best with different hyperparameter settings?
>
> Good point and it is possible. But we still believe it is fair comparison as (1) these hyperparameter settings are inherited from previous work, (2) we didn’t tune them for our dual-mode models, (3) hyperparameter sweep for each model has quite a high cost especially for larger datasets. Thanks for pointing it out and we have added some clarifications in our manuscript.
>
> 4. Visualization of knowledge transfer.
>
> Thanks for the suggestion! We have added a visualization of RNN-T emission lattices of dual-mode models without distillation and models with distillation as a comparison to show why latency is improved. We have updated the manuscript!

---

### Official Review · AnonReviewer2 · 2020-10-28
**An effective, practically relevant approach to unifying end-to-end speech-recognition models for whole-utterance and streaming models**

**Rating:** 7
**Confidence:** 5

**Review:**

The paper proposes a pragmatic approach to unifying end-to-end speech-recognition models for whole-utterance and streaming models. Streaming models are defined as not using any future audio, while whole-utterance (or "full context") models can look at the entire audio recording. In short, the paper shows that multi-task training is effective, where each minibatch uses each utterance twice: Once as using the whole-utterance model structure, and once using a modified model structure where all model weights referring to future information are forced to be zero.

The idea is not revolutionary, but it is interesting to see that it works at no loss of accuracy. Interestingly, the results of the proposed multi-task model are actually a little better compared to training distinct models on the two modalities. This is shown to be the case for two end-to-end model architectures, ContextNet and Conformer. The result is of high practical relevance.

I do disagree with the naming "Universal ASR." The word "universal" is, well, quite universal and can refer to many dimensions, such as whole-utterance vs. streaming, distant vs. close talking, wideband vs. narrowband, various domains, various tasks, multi-linguality, etc. While I understand that in today's crowded ML world, one wants to occupy powerful marketing names for ones own work, I strongly believe that they should not be over-stating. For this paper to be acceptable for publication, I believe the authors should tone down the "Universal" moniker, and choose a different name that delineates the dimension more accurately.

Further, I would like to note that although this is supposed to be a double-blind review, the non-blind reference for the internal corpus gives away that this is a Google paper.

**Quality**: The results have been obtained on two quality data sets, one public generally accepted corpus (LibriSpeech), and one Google-internal set of a whopping 413 thousand hours. I consider the results both convincing and of interest to speech-interested members of the ICLR community.

As one feedback point, the paper does not seem to mention a simple and common technique of incorporating at least some future information, by pre-rolling a fixed number of speech frames, e.g. 5, and incorporating those e.g. by stacking or limited-forward looking lower-layer convolution(s) or self-attention. I think it is important that the authors comment on this, since there is still a one-point WER gap due to streaming. To what degree would a simple lookahead like this trade latency for accuracy? Is it possible to build a model that is universal over different numbers of lookahead frames, as to allow for an adaptive lookahead (e.g. pre-roll 0.5 seconds at sentence start, but less later; maybe depending on context such as the words recognized so far)?

As another feedback point, I find the references somewhat Google-heavy. Going through the references from the start, it takes a while to find a non-Google reference. Exaggerated self-referencing that gives the incorrect impression that Google invented everything is not appreciated by all members of the community. I suggest that the authors rebalance, or at least adjust the order. In some cultures, it is considered polite to mention yourself last.

**Clarity**: The paper generally well-written and easily understandable, although I have a few feedback items.

The term "inplace knowledge distillation" is a little confusing. Initially I thought this was done during inference, and was curious how that may be accomplished. However, it is a training-only technique. Please add that to where this introduced ("at training time").

I kindly ask the authors to be more precise in the results table. What is the unit of Latency? I presume ms, but it could also be speech frames. This is not stated, the string "ms" does not occur once. Also, I find it confusing that the table headings for the WER columns are the corpus names, while for latencies, it is the actual meaning of the column ("Latency@X"). So please (1) use WER for these columns, and (2) add units for both, e.g. by saying "WER [%]" and "Latency@50 [ms]". Further, the readability of Tables 3 and 4 could be improved by horizontally aligning the decimal points.

I also kindly ask the authors to have the text proof-read by a native speaker of English, and/or an automated grammar checker which Google certainly has. I noticed a few missing articles.

**Originality**: With the hindsight of having read the paper, the basic idea does not *seem* original at all, but rather straight-forward. The originality of the paper lies in asking the right question, which in this case then naturally leads to the proposed method.

**Significance**: The method allows to reduce the number of models to train by half, which has tremendous practical impact for anyone operating an ASR service. While there is a clear reduction of overhead of model validation and deployment process, it is not clear from the paper whether training the joint model is cheaper than training two individual models (since each utterance is trained twice per minibatch). It would be great if the authors could share that information somewhere.

**Pros**: An interesting paper that provides a simple, well-working solution to a real practical problem, evaluated on a massive real-life corpus.

**Cons**: Some room for improvement of clarity as listed above. Lack of consideration of simple approach of pre-rolling a few frames of speech.

---

> ### Author Response · Authors · 2020-11-24
> **Authors' Reply to AnonReviewer2**
>
> Thanks for your time reviewing and providing detailed suggestions for our manuscript! We have updated the manuscript according to your suggestions and addressed all concerns below.
>
> ## Part 1: General Questions
>
> Q: Better name instead of “Universal ASR”.
>
> A: Thanks for raising your concern and providing many reasonings. We have re-considered the name and have decided to change it to “Dual-mode ASR'', which is  more accurate and less ambiguous. We agree on your arguments on the name and thanks for your detailed explanations!
>
> Q: Concern on non-blind reference for the internal corpus (MultiDomain dataset).
>
> A: Thanks for raising the concern! The usage of internal data is not rare and has been published many times in previous ICLR conferences, for example, ICLR 2019 Oral Accepted Paper: Large Scale GAN Training for High Fidelity Natural Image Synthesis, uses an internal dataset JFT-300M. We didn’t find the detailed explanation of “double-blind” on ICLR website, but as a peer conference, we found explanation from CVPR website of “Double blind review”: http://cvpr2020.thecvf.com/submission/main-conference/author-guidelines
> And we believe we didn’t violate the rule according to this interpretation.
>
>
> ## Part 2: Detailed Questions / Comments / Suggestions
>
>
> 1. Stacking pre-rolling or limited-forward looking baselines.
>
> Thanks for the suggestion! We added a simple baseline of looking-ahead frames (6 frames, 10ms per frame, totally 60ms look-ahead latency) in Table 3 by padding additional frames at the end of the input utterances. To summarize the result, the look-ahead baseline has better WER but worse emission latency.
>
> 2. More and diverse references.
>
> Thanks for the suggestions! We have revisited the literature and added many more diverse references in our manuscript.
>
> 3. Add “at training time” to inplace knowledge distillation.
>
> Thanks for the suggestions! We have updated the manuscript accordingly.
>
> 4. Unit of Latency? Suggest (1) use WER for these columns, and (2) add units for both, e.g. by saying "WER [%]" and "Latency@50 [ms]".
>
> Our unit of latency is milliseconds and we have updated the draft to include the unit of latency (ms) and WER (%) in all tables. We have also revised the paper writing. Thanks for the suggestion!
>
> 5. Whether training the joint model is cheaper than training two individual models.
>
> A dual-mode ASR model has no more training cost than training both streaming and non-streaming models separately. Moreover, the shared weight part of dual-mode ASR gets supervision signals from both streaming and non-streaming loss, thus the learning is supposed to be more efficient. In practice, we also found joint training converges faster (e.g., LibriSpeech) thus has lower training cost overall. All results in the manuscript are based on fair training cost / number of training iterations and we don’t claim it as a benefit explicitly.

---

### Official Review · AnonReviewer1 · 2020-10-29
**Powerful and practical techniques by unifying streaming and full-context ASR systems**

**Rating:** 7
**Confidence:** 5

**Review:**

This paper proposes a unified single neural network architecture to realize both streaming and full-context ASR systems. The idea is simple but very efficient. It uses the same model for both streaming and full-context ASR systems, but when we use the streaming mode, some of the network operations that use the future context are ignored (e.g., ignore the future context kernel in CNN, self-attention, or pooling is performed only with the current and history frames). The two modes interact with each other by teacher-student training, where the teacher is the full-context mode while the student is the streaming mode. The paper is well-written overall, but it requires some clarification (see my comments below).

One of the drawbacks of this approach is that the current technique is too specific to the ASR topic, and it may not get much attention from the general machine learning audience. The authors mention that we can apply the technique to simultaneous machine translation or video processing. However, I'm not sure because the method seems to depend highly on RNN-T structure (assuming the monotonicity). We could not directly use this for simultaneous machine translation. Also, the paper does not have some novel machine learning algorithms, and it would also not gain much attention from the general machine learning audience. Finally, the result looks very impressive, but it requires some clarifications (see my comments below). Especially, I could not fully understand why this method has better latency than the other methods (sometimes becomes negative). Is that related to some search strategy (end-point detection)? I want to have more clarifications about it.

Other detailed comments:
1. "Universal ASR" is not the correct word to represent the proposed method. Please re-consider this name.
2. The paper claims the state-of-the-art streaming ASR results on the  LibriSpeech task, but I saw some papers (Moritz et al., 2020, Tsunoo et al., 2020) show better performance than this proposed method. Please confirm it and explain more differences. One possible reason is the use of LM. The methods in these papers use an LM, while this paper's method does not seem to use an LM. Please clarify it.
3. "Listen, Attend, and Spell" isa a great study, but the original attention-based encoder-decoder for ASR was proposed by the following papers earlier. The authors should respect these pioneering studies and cite them.
  - Chorowski, Jan, et al. "End-to-end continuous speech recognition using attention-based recurrent NN: First results." NIPS 2014 Workshop on Deep Learning, December 2014. 2014.
  - Chorowski, Jan K., et al. "Attention-based models for speech recognition." Advances in neural information processing systems. 2015.
4. "Weight Sharing for Multi-tasking.": How about discussing Watanabe, Shinji, et al. "Hybrid CTC/attention architecture for end-to-end speech recognition." IEEE Journal of Selected Topics in Signal Processing 11.8 (2017): 1240-1253? Although the motivation is different, the methodology is very similar.
5. Section 3 is well summarized. It well describes dual modes' design by making a clear distinction between the full-context and streaming operations.
6. How much is more/less training time required when we use the dual-mode than single-mode or separately training both modes?
7. "Latency@50 and Latency@90": I think it's reasonable to exclude the outlier result, but at the same time, this outlier would be more critical for the user satisfaction perspective. Thus, I'm curious about the outlier, and it would be great if the authors discuss the outlier more.
7. I could not find the detailed machine spec information when computing the latency. Please clarify it.
7. Section 4.1 has a clear relationship with the discussion in Section 3, and it is easy to follow.
8. The table should have the unit for the latency. Millisecond?
9. Compared with the other prior studies (Moritz et al., 2020), the streaming mode's degradation from the full-context mode seems to be larger. Please explain it.

---

> ### Author Response · Authors · 2020-11-24
> **Authors' Reply to AnonReviewer1**
>
> Thanks for your time reviewing and providing detailed suggestions for our manuscript! We have updated the manuscript according to your suggestions and addressed all concerns below.
>
> ## Part 1: General Questions
>
> Q: Paper is too specific to the ASR, not for the general machine learning audience.
>
> A: It is true that our main focus in this paper is to unify streaming and non-streaming ASR, as speech recognition is one of the major topics listed on ICLR conference website. We also fully agree it would be nice if the technique can be modified and adapted to simultaneous machine translation or video processing. We leave them as future work in our Conclusion.
>
> Q: Paper does not have some novel machine learning algorithms.
>
> A: While novelty can have many forms, we believe our exploration of unifying streaming and full-context ASR models and the presented experiments are novel and motivating. The techniques proposed in this work could open up many new possibilities of research directions and applications as explained in Introduction and Conclusion.
>
> Q: This method has better latency than other methods, is it because of end-point detection.
>
> A: The improvement of latency is not because of endpointing, as all of our baselines have the same endpointing methods and benchmark metrics. As shown in Ablation Study Table 4, since non-streaming ASR models do not have latency, we found joint training streaming ASR and non-streaming ASR together with weight sharing and distillation enforces streaming ASR models to intuitively “mimic” non-streaming models thus emit faster with lower latency. We have also added a visual comparison of Dual-mode ASR models with and without knowledge distillation to better show visually how the latency can be improved.
>
>
> ## Part 2: Detailed Questions / Comments / Suggestions
>
> 1. Re-consider name “Universal ASR”.
>
> Thanks for raising the concern of naming. We have re-considered the name “Universal ASR” and have decided to change it to “Dual-mode ASR”. We hope “Dual-mode ASR” can be more accurate and less ambiguous. Thanks for the suggestion!
>
> 2. Confirm LM is not used for LibriSpeech results.
>
> Yes we confirm we do NOT use LM for our LibriSpeech results. Moreover, baselines (Moritz et al., 2020, Tsunoo et al., 2020) have much higher latency (e.g., best streaming WER achieved in Moritz et al., 2020 has latency 2190ms, while our averaged latency is ~50ms). If we compare models of similar latency (regardless of LM), our WERs are better.
>
> 3. Cite pioneering studies.
>
> Sure we are more than happy to cite and discuss pioneering research work and thanks for your reference! We have updated the manuscript in Related Work.
>
> 4. Related work on Weight Sharing for Multi-tasking.
>
> Thanks again for the reference! We have updated the manuscript and discussed it in Related Work.
>
> 5. Section 3 is well summarized.
>
> Thanks for your acknowledgement!
>
> 6. How much is more/less training time required when we use the dual-mode than single-mode or separately training both modes.
>
> A dual-mode ASR model has no more training cost than training both streaming and non-streaming models separately. Moreover, the shared weight part of dual-mode ASR gets supervision signals from both streaming and non-streaming loss, thus the learning is supposed to be more efficient. In practice, we also found joint training converges faster (e.g., LibriSpeech) thus has lower training cost overall.
>
> 7. Outliers would be more critical for the user satisfaction perspective.
>
> Latency@50 and Latency@90 roughly mean “50% of the users” or “90% of the users”are satisfied which represents the majority, thus are more widely used for user satisfaction study. We agree outliers might be also important and we did find Latency@Average is quite close to Latency@50 in both datasets presented in this paper. For example, Universal Conformer on LibriSpeech (Table 3) has Latency@50 as 10ms and Latency@Average as 11.3ms.
>
> 8. Detailed machine spec for latency.
>
> Our reported latency is model emission latency which is an internal property of the model. To eliminate the difference of machine spec, we did not measure model runtime latency following many previous works (Chang et. al; 2020, Sainath et. al; 2020, Li et. al; 2020).
>
> 9. Section 4.1 is well-written.
>
> Thanks for your acknowledgement!
>
> 10. Unit for the latency. Millisecond?
>
> Yes it is millisecond. We have updated the Tables. Thanks for the comment!
>
> 11. Compared with the other prior studies (Moritz et al., 2020), the streaming mode's degradation from the full-context mode seems to be larger.
>
> Good observation! As also explained in comment (2), best streaming WER achieved in Moritz et al., 2020 has latency 2190ms, while our averaged latency is ~50ms. With much higher latency, the streaming results can be closer to full-context results. But in real-world applications, high-latency streaming ASR is less useful and we aim at low-latency streaming ASR results in this work.

---

### Author Response · Authors · 2020-11-24
**Title Change from "Universal ASR" to "Dual-mode ASR" to inform ACs and all Reviewers**

Dear Readers, Reviewers and Area Chairs,

We would like to thank you again for your time reading, reviewing and providing detailed suggestions for our manuscript!

Thanks to the suggestions from AnonReviewer1 and AnonReviewer2, we have re-considered the name “Universal ASR” and have decided to change it to “Dual-mode ASR”. We hope “Dual-mode ASR” can be more accurate and less ambiguous, as also suggested by AnonReviewer1 and AnonReviewer2. Beyond, we have also revised the manuscript to address the concerns from all reviewers. All our revisions have followed ICLR author's guide.

Best wishes and stay safe,

Authors of ICLR 2021 Conference Paper103

---

### Decision · Program_Chairs · 2021-01-07
**Final Decision**

**Decision:**

Accept (Poster)

**Comment:**

This paper proposes an approach to unifying both full-context and streaming ASR in a single end-to-end model.   Techniques such as weight sharing, joint training and teacher-student knowledge distillation are used to improve the training.  The so-called dual-mode ASR is evaluated under the ContextNet and Conformer networks on Librispeech and MultiDomain datasets. The performance is good.  While the technical novelty is not overwhelmingly significant, all reviewers agree that it may have impact to the speech machine learning community as high-performance streaming ASR is of great importance in real-world deployment of ASR systems.  The authors have meticulously addressed the reviewers' comments and, in particular, changed the title from "universal ASR" to "dual-mode ASR" as suggested by some of the reviewers.  After the rebuttal, all reviewers are supportive on accepting the paper.